# Pediatric Chest Pain: A Review of Diagnostic Tools in the Pediatric Emergency Department

**DOI:** 10.3390/diagnostics14050526

**Published:** 2024-03-01

**Authors:** Szu-Wei Huang, Ying-Kuo Liu

**Affiliations:** 1Emergency Department, Wan Fang Hospital, Taipei Medical University, Taipei 11695, Taiwan; sean04091987@gmail.com; 2Department of Pediatrics, Wan Fang Hospital, Taipei Medical University, Taipei 11695, Taiwan

**Keywords:** chest pain, pediatric chest pain, diagnosis, ultrasound, echocardiography, X-ray, electrocardiogram

## Abstract

Pediatric chest pain is a common chief complaint in the emergency department. Not surprisingly, children with chest pain are usually brought to the emergency department by their parents out of fear of heart disease. However, chest pain in the pediatric population is generally a benign disease. In this review, we have identified musculoskeletal pain as the most prevalent etiology of chest pain in the pediatric population, accounting for 38.7–86.3% of cases, followed by pulmonary (1.8–12.8%), gastrointestinal (0.3–9.3%), psychogenic (5.1–83.6%), and cardiac chest pain (0.3–8.0%). Various diagnostic procedures are commonly used in the emergency department for cardiac chest pain, including electrocardiogram (ECG), chest radiography, cardiac troponin examination, and echocardiography. However, these examinations demonstrate limited sensitivity in identifying cardiac etiologies, with sensitivities ranging from 0 to 17.8% for ECG and 11.0 to 17.2% for chest radiography. To avoid the overuse of these diagnostic tools, a well-designed standardized algorithm for pediatric chest pain could decrease unnecessary examination without missing severe diseases.

## 1. Introduction

Chest pain is a common chief complaint in the emergency department in both adult and pediatric populations. In adults, several life-threatening cardiac diseases might present as chest pain, such as acute myocardial infarction, aortic dissection, arrhythmia, and sudden cardiac death. The idea that “chest pain is related to cardiac disease” is ingrained in the general population. In addition, this idea extends to the pediatric population. Children with chest pain are usually brought to the emergency department by their parents out of fear of heart disease [1,2,3,4,5,6]. Therefore, various examinations might be performed in the emergency department, including electrocardiogram (ECG), chest radiogram, cardiac enzymes such as troponin T/I, and echocardiography. However, chest pain in the pediatric population is generally a benign disease, with the most common cause being musculoskeletal pain, which can be identified simply with detailed history-taking and physical examination [1,3,4,7,8]. Cardiac chest pains in children, on the other hand, are rare, accounting for 0.3–8.0% of all children and adolescents with chest pain (Table 1) [1,2,4,7,9,10,11,12,13,14,15,16,17,18,19]. Several studies have noted that many examinations performed in the emergency department have limited diagnostic value [1,2,4,9,20,21]. For adults, there are several scoring systems to predict major adverse cardiac events (MACE) in patients with acute chest pain, such as the thrombolysis in myocardial infarction (TIMI) risk score, the global registry of acute coronary events (GARCE) score, and the HEART score. These scoring systems are widely used in emergency departments, and each of them revealed different sensitivity and specificity to MACE [22,23,24,25,26,27,28,29]. In pediatric patients, a scoring system is not well-established.

On the other hand, previous studies have demonstrated that standardized algorithms for pediatric chest pain could decrease unnecessary examinations without missing potentially severe diseases [7,30]. In this review, we aim to review the etiology of chest pain in the pediatric population and the common examinations used in the emergency department. However, the utilization of diagnostic tools was relevant to the etiology of pediatric chest pain. For instance, cardiac troponin was frequently used in pediatric patients in the emergency department to identify potentially life-threatening conditions, such as myocarditis. Consequently, an etiological review was conducted as part of this study.

## 2. Methods

To collect relevant studies, previous publications were searched in PubMed, Google Schooler, Clinical Keys, and Springer using keywords and combined word searches (pediatric chest pain, chest pain, emergency department, ECG, X-ray, chest radiogram, ultrasound, echocardiography, heart echo, cardiac ultrasound) for the period from 1 September 1990 to 31 December 2023. Study types including prospective cohort studies, retrospective cohort studies, population studies, meta-analyses, and observational studies were eligible for inclusion. While searching for studies, both authors conducted abstract screening. Abstracts mentioning the etiology of pediatric chest pain or diagnostic tools for pediatric chest pain, including ECG, X-ray, chest radiogram, ultrasound, echocardiography, heart echo, and cardiac ultrasound, were selected for final data extraction. Studies focused on chest pain in an exclusive adult population (age > 18 years old) were excluded.

## 3. Etiology of Chest Pain in Children and Adolescents

Chest pain in children and adolescents could be divided into the following etiologies: musculoskeletal, idiopathic, pulmonary, gastrointestinal, psychogenic, cardiac, or miscellaneous causes [1,2,9,12,13,14,15,16,17,18,19]. Due to varying classifications of the causes of chest pain in different articles and the lack of uniformity in diagnostic definitions among these articles, the exact proportion of chest pain in children and adolescents varies from study to study. However, it is clear that cardiac chest pain is relatively rare in the pediatric population, accounting for 0.6–8.0% [1,2,9,12,13,14,15,16,17,18,19].

Most of the studies found that the causes of chest pain in children and adolescents are predominantly musculoskeletal and/or idiopathic, accounting for 38.7–86.3% [1,2,9,12,13,14,15,16,17,18,19]. Chest pain due to musculoskeletal causes can often be diagnosed through medical history and physical examination alone, with limited assistance from additional diagnostic examinations. Common symptoms and physical examination findings in musculoskeletal chest pain include tenderness in the chest wall, pain provoked by muscle contraction, or pain with movement. Musculoskeletal and/or idiopathic chest pain are mostly self-limiting and spontaneously resolve over time without specific treatment [14].

Pulmonary system-related chest pain is also not highly prevalent, accounting for 1.82–12.78% [1,9,12,14,15,16,17,31], with common causes including pneumothorax, asthma, pneumonia, and the less common pneumomediastinum [32,33,34,35,36,37,38,39]. Some of these diseases can be potentially life-threatening, such as tension pneumothorax. However, these respiratory diseases can be identified through simple physical examination: diminished breath sounds in pneumothorax, wheezing in asthma, respiratory infection symptoms, abnormal breath sounds in pneumonia, etc. [40,41,42,43,44]. While X-rays can assist in diagnosing these conditions, a thorough medical history and physical examination are sufficient for initial exclusion.

Gastrointestinal chest pain is also relatively rare in children and adolescents, comprising about 0.2–9.3% [1,2,12,14,15,16,18]. Gastrointestinal chest pain could be caused by gastroesophageal reflux, gastric ulcers, or gallbladder diseases. Common tests for chest pain, such as electrocardiograms, X-rays, echocardiography, and blood examinations, yield limited assistance. On the other hand, such chest pain often has a relative history, including meal-related pain, burning sensation in the chest, gastroesophageal reflux, difficulty swallowing, or vomiting [14]. Therefore, in differentiating chest pain caused by gastrointestinal etiology in children and adolescents, history and physical examination are important.

Psychogenic chest pain is also a common cause among children and adolescents. According to previous studies, it accounts for about 5.1–83.6% of all cases in this group [1,2,9,14,17,18,19]. Recent studies have found that psychogenic chest pain in children and adolescents may exceed 20.0% [1,14,17,19], with some studies indicating as high as 83.6% [17]. Rita Pissarra et al., in a study of 798 children and adolescents with chest pain, found that 21.6% had psychogenic chest pain, with risk factors including being female, adolescence, previous mental illness, previous high-stress events, and normal physical examination [1]. On the other hand, Emre Aygun et al. found that 28.4% of chest pain in children and adolescents was psychogenic and more prevalent in those older than 12 years [19]. However, in contrast to Rita Pissarra et al., Emre Aygun et al. reported that psychogenic chest pain was more common in males [19]. With increased attention to children and adolescents’ mental health in recent years, diagnoses of psychogenic chest pain have begun to rise [1,14,17,19,45]. However, due to the lack of a unified assessment method and diagnostic criteria, the true proportion of psychogenic chest pain and management remains unclear. Future research should focus more on psychogenic chest pain in children. Common examinations for chest pain in children and adolescents yield limited value in diagnosing psychogenic chest pain. However, with the increasing incidence of psychogenic chest pain, it should be included in the differential diagnosis when evaluating children and adolescents with chest pain.

Finally, cardiac chest pain is the most concerning issue for patients, parents, and healthcare professionals. In previous studies, cardiac chest pain in children and adolescents accounted for about 0.3–8.0% [1,2,9,12,13,14,15,16,17,18,19]. This wide range is mainly due to different definitions of cardiac chest pain used in the studies. In most studies, chest pain in patients with myocardial infarction, coronary artery anomalies, congenital structural heart disease, myocarditis, pericarditis, arrhythmias (Wolff–Parkinson–White syndrome, premature atrial contraction, premature ventricular contraction, tachyarrhythmia), Kawasaki disease, dilated cardiomyopathy (DCM), and hypertrophic cardiomyopathy (HCM), is classified as cardiac chest pain, with a proportion of 0.6–1.8% (Table 2) [1,9,12,13,17]. On the other hand, some studies include suspected myocarditis or mitral valve prolapse in the category of cardiac chest pain, raising its proportion to 6.8–8.0% [15,16,18]. Due to the significant heterogeneity in past research and the lack of a universal definition of cardiac chest pain, the true proportion of chest pain in children and adolescents remains unclear. However, it is known that cardiac chest pain is not highly prevalent in the pediatric population. Therefore, clinicians need to be more cautious in their diagnostic approach to avoid overuse of examination tools.

## 4. Common Examinations Performed for Pediatric Chest Pain in the Emergency Department

### 4.1. Electrocardiogram

An electrocardiogram (ECG) is a noninvasive examination that can be performed within a few minutes. Several critical cardiac diseases can be detected using an ECG, including acute myocardial infarction, myocarditis, and fatal arrhythmia. Therefore, in many institutions, ECG is a standard examination for adult patients with the chief complaint of acute chest pain. In contrast to adults, cardiac chest pain is rare in children and adolescents. In previous studies, ECG was performed in 27.8–62.4% of pediatric patients with the chief complaint of chest pain, with a sensitivity of 0–17.8% [1,3,9,11,12,13,46]. In addition, the actual specific ECG finding for cardiac etiology might be even lower because of the difference in the definition of “positive ECG findings” used in the studies. In some studies, positive ECG was defined as any abnormalities, such as sinus tachycardia/bradycardia, abnormal repolarization, preexcitation, and incomplete bundle branch block [9,10,11]. In a retrospective chart review of 16,147 pediatric ECGs performed in the pediatric emergency department, Theiler et al. [47] divided ECG findings into three classes. Class I was defined as mild abnormalities for which no cardiology follow-up was needed, class II was defined as moderate abnormalities for which cardiology follow-up was recommended, and class III was defined as severe abnormalities for which immediate intervention was warranted. Class III abnormalities included third-degree AV block, atrial tachycardia, supraventricular tachycardia, ST elevation/depression, and wide QRS tachycardia. The authors found that class III ECG abnormalities were present in only 2.0% of children and adolescents who underwent ECG at the emergency department [47]. Due to differences in the definitions of positive ECGs, it is very difficult to determine the actual sensitivity and specificity of ECGs for cardiac chest pain.

On the other hand, Pissarra et al. [1] found that in nine pediatric patients with cardiac chest pain, only 77.7% of the patients showed ECG alterations. These nine patients had a final diagnosis of two cases of arrhythmia and seven cases of myocarditis/pericarditis/myopericarditis.

Due to the above characteristics of ECG in the pediatric population, ECG is not an efficient tool to be used as a general screening tool for chest pain of cardiac etiology. In the previously mentioned ECG study, Theiler et al. [47] found that the indication of cardiac concern, electrolyte abnormalities, and emergent or critical triage level had increased odds of clinically significant ECG compared with the indication of chest pain only. Similarly, in a retrospective ECG review in the pediatric emergency department, Gandhi et al. [10] found that the chief complaint of chest pain in the pediatric population in emergency department had a lower odds ratio (OR) (OR: 0.383, CI: 95% = 0.18–0.80) compared with clinical findings of tachypnea (OR: 1.74, CI: 95% = 1.42–2.62), tachycardia (OR: 1.85, CI: 95% = 1.10–3.09) or bradycardia (OR: 3.69, CI: 95% = 1.47–9.28). Mohan et al. [3] found that a well-designed clinical pathway for pediatric chest pain in the emergency department did not miss a diagnosis of cardiac chest pain, with an ECG usage of 47.7% among all patients with chest pain.

Despite the convenience of ECG, emergency health providers should know that there are rare cardiac etiologies in pediatric chest pain, and ECG should be used in select patients with a higher chance of having cardiac chest pain, such as patients with tachypnea, tachycardia, bradycardia, electrolyte abnormalities, and emergent or critical triage levels.

### 4.2. Chest Radiography

Chest radiography is also a common examination for children and adolescents with acute chest pain. In previous studies, 35.6–72.0% of pediatric patients with chest pain underwent chest radiography [1,9,12,13,33]. Common positive findings on chest radiography included pneumonia/bronchitis, pneumomediastinum, pneumothorax, pneumopericardium, and cardiomegaly. However, the sensitivity of chest radiography was only 11.0–17.2% [1,9,12,13,33].

Pneumothorax and pneumomediastinum are two of the diseases that emergency physicians want to identify via chest radiology. However, pneumothorax and pneumomediastinum accounted for only 3% of pediatric chest pain cases of pediatric emergency department visits [4]. Rowe et al. [9] found that in pediatric patients with chest pain who underwent chest radiography in the emergency department, only 1.9% of patients had pneumothorax or pneumomediastinum. Similarly, Pissarra et al. [1] found that in 420 radiographs of pediatric patients with chest pain, only 2.4% were diagnosed with pneumothorax or pneumomediastinum.

Pneumonia is also a common pediatric pulmonary disease that might present with chest pain. Several studies showed that in children and adolescents with the chief complaint of chest pain, pneumonia was identified in 3.7–9.3% of all patients.

Although several diseases presenting with chest pain in the pediatric population can be identified via chest radiography, we should know that the sensitivity of chest radiography is low, and many of these diseases could also be identified with careful history-taking and physical examination. Pissarra et al. [1] found that in pediatric patients with pulmonary chest pain, there was a significant association with respiratory antecedents and pulmonary auscultation alteration. In addition, Mohan et al. [3] also found that a clinical pathway for chest pain in the pediatric department could significantly decrease the use of chest radiography from 46.1% to 35.6%.

Similar to ECG, chest radiography might be useful in identifying several diseases presenting with chest pain in children and adolescents. Due to the relatively low positive ratio, using a clinical pathway for pediatric chest pain, careful history-taking, and physical examination might decrease the unnecessary use and enhance the positive ratio of chest radiography.

### 4.3. Echocardiogram

For further evaluation of chest pain, an echocardiogram could be used to identify cardiac etiology. Ventricular systolic function, pericardial effusion, and anatomical abnormalities could be evaluated through an echocardiogram. However, it must be performed by well-trained cardiologists or medical providers. Therefore, it is reasonable that echocardiography was not a commonly performed examination at the pediatric emergency department for patients with chest pain. In recent studies, only 1–1.5% of pediatric patients with acute chest pain in the pediatric emergency department underwent an echocardiogram [3,13,48,49]. Despite being rarely used, Mohan et al. [3] found that no cases of severe cardiac diseases were missed in a retrospective review. In addition, Drossner et al. [13] found a significantly higher use of echocardiogram in patients with cardiac-related chest pain compared with patients with noncardiac chest pain, suggesting that there is still diagnostic value of the echocardiogram. The diseases caused by cardiac-related chest pain included pericarditis, myocarditis, myocardial infarction, supraventricular tachycardia, long QT syndrome, and ventricular tachycardia [13]. The decision to arrange an echocardiogram in pediatric patients with chest pain at the emergency department lacks a universal guide. Most of these decisions were made according to the clinical judgment of emergency physicians. We should know that echocardiograms have a certain diagnostic value for cardiac-related chest pain, and they might be effective in a relatively low frequency of use.

On the other hand, cardiac points-of-care ultrasound (POCUS) is a useful tool to identify global systolic dysfunction and pericardial effusion, with a sensitivity of 100% and a specificity of 99.5% [21]. Miller et al. [21] also found that in selected pediatric patients with cardiac-related complaints, such as chest pain, dyspnea, and tachycardia, the prevalence of pericardial effusion was 11% and the prevalence of global systolic dysfunction was 4%, which could be identified by cardiac POCUS. This information might be helpful when certain cardiac diseases are suspected, such as myocarditis or pericarditis.

Echocardiography is a useful tool to identify certain cardiac diseases. On the other hand, cardiac chest pain is rare in the pediatric population; therefore, it is important to use echocardiograms wisely without overuse. In addition, cardiac POCUS could identify global systolic function and pericardial effusion with high validity, and it is also a timely examination that could be rapidly performed in the pediatric emergency department. Both echocardiogram and cardiac POCUS could be used as diagnostic tools for selected children and adolescents with chest pain to identify cardiac abnormalities, including pericardial effusion and systolic dysfunction.

### 4.4. Troponin

Cardiac troponin is a highly specific and sensitive laboratory parameter for detecting myocardial injury. In adults, cardiac troponin is often used to detect acute myocardial infarction in the emergency department [50]. On the other hand, cardiac troponin is also used in children and adolescents who complain of acute chest pain in the emergency department to detect cardiac etiology, including myocarditis and pericarditis. In pediatric patients with chest pain in the emergency department, cardiac troponin was performed in 2.6–8.9% of all patients [1,3,51,52,53]. In a study of pediatric chest pain in the emergency department, Brown et al. [51] found that in pediatric patients who underwent cardiac troponin examination, 17.5% of patients had elevation, and the majority of these patients (48%) had a final diagnosis of myocarditis or pericarditis.

Although cardiac troponin is a sensitive biomarker for myocardial injury and myocarditis/pericarditis are potentially life-threatening diseases in the pediatric population, there are still several pieces of evidence we should know. First, there are still many noncardiac etiologies that could cause the elevation of cardiac troponin in the pediatric population. In an 11-year period study, Yoldaş et al. [52] found that in children and adolescents with elevated cardiac troponin, more than half of the cases (53.2%) were caused by noncardiac etiology, which included drug intoxication, carbon monoxide intoxication, bronchopneumonia–asthma, shock, sepsis, status epilepticus, and asphyxia. In patients with cardiac etiologies, the majority were still myocarditis/pericarditis (64%), followed by cardiomyopathy and arrhythmia [18]. Assandro et al. [54] also pointed out that troponin is a sensitive but not specific marker for screening pediatric myocarditis, and it should be used only if cardiac etiology is highly suggestive and after an electrocardiogram and chest radiography. Second, cardiac troponin could be normal in patients with myocarditis or pericarditis. In a review article on chest pain in pediatric patients, Jennifer Thull-Friedman et al. [7] found that in patients with myocarditis, only 54% of patients had troponin elevation and 73% of patients had creatine kinase elevation. On the other hand, in patients with myocarditis, up to 93–100% of the patients had abnormal electrocardiogram [4]. Due to its low sensitivity and low specificity to myocarditis, cardiac troponin should not be used as a single tool for screening myocarditis. The clinical diagnosis of myocarditis should be made by comprehensive evaluation, including history-taking, clinical signs and symptoms, ECG, echocardiography, and X-ray, as well as cardiac troponin [51,52,53,54,55,56,57,58,59,60,61,62,63,64,65]. Lastly, despite being rare in the pediatric population, acute coronary syndrome can still occur. Brown et al. found that in 212 pediatric patients with chest pain, 6 of them (2.8%) had a diagnosis of acute myocardial infarction, and cardiac catheterization was promptly arranged [51]. Therefore, acute myocardial infarction, although rare in the pediatric population, should be put into differential diagnosis in patients with acute chest pain and troponin elevation.

Cardiac troponin is a common laboratory test used to detect myocardial injury, and it is often used in pediatric patients with complaints of chest pain, syncope, and arrhythmia [52]. Although troponin could be helpful in diagnosing myocarditis and pericarditis in the pediatric population, we should know that there are still many diseases that could cause troponin elevation. Therefore, cardiac troponin is not an ideal tool for screening pediatric patients with chest pain for cardiac etiology, but it is a useful tool for detecting cardiac disease in selected patients under the suspicion of heart disease, namely, those with a clinically suspicious history of cardiac etiology, such as alteration of physical examination, abnormal ECG fining, or abnormal chest radiogram.

### 4.5. Other Examination

Other examinations were arranged in pediatric patients with acute chest pain, such as blood analysis and urine analysis. In different studies, 5.6–26.5% of children and adolescents with chest pain in the emergency department underwent these examinations, and 9–19.2% of the patients had abnormalities [1,3,9,12,13]. The diagnostic value for pediatric chest pain was very difficult to evaluate because various examinations were performed. However, the etiology of chest pain in pediatric chest pain includes musculoskeletal pain, idiopathic chest pain, pulmonary chest pain, gastrointestinal-related chest pain, psychogenic chest pain, and cardiac chest pain [1,2,3,4,10,11,12,13]. In the studies, musculoskeletal pain and idiopathic chest pain were the majority causes of pediatric chest pain, followed by psychogenic chest pain, pulmonary-related, and gastrointestinal-related [1,3,4,9,12]. Many of these diseases could be differentiated through detailed history-taking and physical examination. Other examinations, such as blood examination or urine analysis, yield limited diagnostic value in these diseases.

However, this does not suggest that these examinations are not necessary in pediatric patients with chest pain, but they should be used in selected patients. Drossner et al. [13] found that there was significantly higher use of laboratory tests in pediatric patients with cardiac-related chest pain than in those with noncardiac chest pain (79% vs. 26%). The final diagnosis included pericarditis, myocarditis, myocardial infarction, supraventricular tachycardia, long QT syndrome, ventricular tachycardia, pulmonary embolism, and pneumopericardium. Therefore, similar to cardiac troponin, these examinations might also be helpful if certain diseases are suspected but are not useful as a universal screening tool for pediatric chest pain.

### 4.6. Discussion—Cardiac Chest Pain in the Pediatric Population

Cardiac chest pain is a frequently encountered complaint in both outpatient and emergency settings among children, often driving parents and patients to associate it with heart-related diseases, leading them to seek medical assistance [1,2,3,4,5,6]. Conversely, healthcare professionals, aiming to rule out the cardiac origin of chest pain, often conduct various diagnostic tests, including electrocardiography, X-rays, echocardiography, and cardiac enzyme assays. However, these tests exhibit suboptimal sensitivity for identifying cardiac chest pain and are not suitable as screening tools [1,2,3,9,10,12,13,47,51].

The prevalence of cardiac chest pain in children is approximately 0.3–8.0%, with varying proportions depending on different study designs and definitions of cardiac chest pain. Common diseases contributing to chest pain include myocarditis/pericarditis, structural heart diseases, arrhythmias, cardiomyopathy, myocardial infarction, and others [1,9,12,14,16,17,18].

Myocarditis/pericarditis is one of the most severe diseases for emergency doctors due to its rapid progression and high mortality rates. However, only 0.2–1.8% of chest pain cases in children and adolescents are attributed to myocarditis/pericarditis (Table 2). Furthermore, there is no single highly specific diagnostic tool for acute myocarditis/pericarditis. A diagnosis of myocarditis/pericarditis could be made via comprehensive evaluations, which include clinical history, signs, symptoms, various tests, and examinations [62,66,67,68,69]. Previous studies have shown that symptoms such as gastrointestinal manifestations (abdominal pain, vomiting, decreased appetite), circulatory compromise such as low blood pressure, and respiratory distress are common symptoms and signs in pediatric myocarditis [65,70]. Targeted examinations for patients presenting with these symptoms and signs can enhance diagnostic sensitivity, thus reducing the unnecessary use of diagnostic tools.

Previously undiagnosed structural heart diseases in children and adolescents with chest pain are relatively rare, accounting for only 0.3–1.1% (Table 2). These diseases, including atrial and ventricular septal defects, coronary artery anomalies, and aortic and pulmonary stenosis [66,67,71,72,73], often present with abnormal heart sounds detectable during physical examinations such as systolic murmur in patients with ventricular septal defect, aortic and pulmonary stenosis, or fixed S2 split in patients with atrial septal defect [74,75,76,77,78,79]. Detailed auscultation and physical examinations can provide the initial exclusion of these diseases, supplemented by echocardiography for definitive diagnosis.

Arrhythmias or abnormal electrocardiograms (ECGs) are uncommon in pediatric and adolescent chest pain, constituting only 0.14–2.7% (Table 2) [66,67,71,72,73]. Previous studies showed that an ECG examination was ordered in approximately 27.8–62.4% of children and adolescents with chest pain, with 10–17.8% of them testing positive [1,3,9,10,12,13,47]. On the other hand, Shiv Gandhi et al. found that in children and adolescents with chest pain, the likelihood of abnormal ECG results is higher when the following conditions are present: underlying heart diseases, the presence of symptoms of tachycardia, and concurrent symptoms of dyspnea/tachypnea [10]. Similarly, in a retrospective cohort study of 504 pediatric patients with arrhythmia transported by Helicopter Emergency Medical Service, 82.7% had supraventricular tachycardia (SVT), 6.7% had sinus bradycardia, 2.1% had ventricular tachycardia, 2.1% had atrial tachycardia, and 1.9% had AV block [80]. This highlights tachyarrhythmias as the predominant rhythm among children, identifiable initially through auscultation. Due to its non-invasive nature and rapid assessment, an ECG is frequently ordered in emergency settings. However, physicians should be aware that arrhythmias or ECG abnormalities causing chest pain are not commonly encountered in this population. Performing ECGs for specific chest pain presentations, such as those with known heart disease, symptoms of tachycardia, and concomitant respiratory distress, can increase diagnostic sensitivity [10].

Cardiomyopathy accounts for a small proportion (0.05–0.2%) of chest pain cases in children and adolescents [15,16,18]. Diagnosis of cardiomyopathies requires specialized evaluations by cardiologists, including ultrasound, magnetic resonance imaging, computed tomography, cardiac catheterization, and genetic testing [81,82,83]. Given the difficulty of immediate diagnosis in the emergency setting, factors influencing the disposition of these patients should be based on clinical symptoms, including vital signs, signs and symptoms of heart failure, and respiratory symptoms such as tachypnea and dyspnea.

Acute coronary syndrome is a common emergency in adult chest pain patients [84,85,86]; however, it is rarely seen in pediatric chest pain, accounting for only 0.06% [13]. In addition, in many large-scale studies on chest pain in the pediatric population, no acute coronary syndrome was diagnosed [1,15,16,18]. On the other hand, clinical presentations of acute coronary syndrome could be very similar to myocarditis/pericarditis in children and adolescents [87,88,89,90,91]. Therefore, similar to the approach for patients with myocarditis/pericarditis, the diagnosis of coronary artery disease in children and adolescents requires a comprehensive evaluation based on clinical symptoms and various tests.

Other causes of cardiac chest pain contribute to approximately 0.04–6.2% of cases, which include suspected myocarditis, mitral valve prolapse, pulmonary embolism, pneumopericardium, pulmonary hypertension, Marfan syndrome, and Kawasaki disease [13,15,16,18]. Li Chen et al. found that, among these etiologies, the majority cause of cardiac chest pain was suspected myocarditis, accounting for 93.6–96.2% of cases [15,16]. On the other hand, Muhammed Karabulut et al. found that among cardiac chest pain cases due to other causes, patients often had mitral valve prolapse. Other causes of cardiac chest pain, such as pulmonary embolism, pneumopericardium, pulmonary hypertension, Marfan syndrome, and Kawasaki disease, represent sporadic cases in various studies [13,15,16,18]. Due to the relative rarity of these conditions and the limited sensitivity of commonly used emergency diagnostic tools, alternative diagnostic methods should be used for differentiating these diseases.

Finally, many studies have also indicated that well-designed clinical pathways for chest pain in children and adolescents can effectively reduce the misuse of diagnostic resources without compromising accuracy in diagnosis [5,7,30,92]. Shaun Mohan et al. found that the implementation of a well-designed chest pain diagnostic pathway in the emergency department does not increase the misdiagnosis of cardiac chest pain. Moreover, it decreases the use of X-rays and may increase outpatient follow-up rates [3]. In a cohort study, Kevin G. Friedman et al. also discovered that the utilization of a standard chest pain assessment pathway can reduce the use of cardiac ultrasound and outpatient rhythm monitors by 20% while still effectively diagnosing all cases of cardiac chest pain [7].

### 4.7. Special Consideration: Chest Pain in Adolescents with Marfan Syndrome

Marfan syndrome, an autosomal dominant disease resulting from mutations in the FBN1 gene, is predominantly recognized for its cardiac vascular manifestations, including cardiac valvular diseases such as aortic regurgitation and mitral valve prolapse, aortic root dilatation, and life-threatening complications such as aortic dissection [93,94,95,96,97,98,99,100,101,102]. On the other hand, aortic dissection also stands as one of the differential diagnoses in patients presenting with acute chest pain. Prompt consultation and guidance from cardiac surgeons are imperative upon establishing a diagnosis [93]. Consequently, aortic dissection is one of the most important differential diagnoses for emergency doctors when facing a patient with acute chest pain. Previous research indicates that aortic dissection in patients with Marfan syndrome predominantly occurs after adolescence [97,99,100,101]. Hascoet et al., in a cohort study involving 462 patients with Marfan syndrome, identified four cases of aortic dissection. The ages of these four patients were 15.0, 15.9, 24.6, and 24.9 years, respectively [99]. Similarly, Wozniak-Mielczarek et al., in a study involving 101 pediatric and adult patients with Marfan syndrome, found that 8 patients (7.92%) experienced aortic dissection, with 2 pediatric patients aged 16 and 17 years, respectively [101]. On the other hand, aortic dissection in children with Marfan syndrome is sparse [103,104,105].

Given that aortic root dilatation is a characteristic feature of Marfan syndrome, the incidence of aortic dissection increases with an enlarged size of the aortic root [93,98]. Hence, it is reasonable that adolescents with Marfan syndrome are more prone to aortic dissection compared with children. Although aortic dissection is rare in children and adolescents [97,99,100,101,103,104,105], individuals with Marfan syndrome inherently carry a risk of aortic root dilatation, predisposing them to aortic dissection. Therefore, considering aortic dissection in the differential diagnosis and conducting further investigations in adolescents with Marfan syndrome suffering from acute chest pain is reasonable.

### 4.8. Special Consideration: Children and Adolescents with Chest Pain Following COVID-19 mRNA Vaccination

Following the outbreak of the COVID-19 pandemic at the end of 2019, mRNA vaccines, such as BNT162b2 mRNA-Pfizer-BioNTech, and mRNA-1273-Moderna, were introduced by the end of 2020, with administration to adolescents commencing from May 2021. While COVID-19 mRNA vaccines are considered highly safe and effective in preventing COVID-19 transmission and severe complications, concerns persist regarding potential adverse effects, particularly myocarditis and pericarditis. Pediatric and adolescent individuals manifesting myocarditis or pericarditis after COVID-19 mRNA vaccination frequently exhibit chest pain (93.3–100.0%), prompting heightened concern among themselves or their families, leading to the pursuit of medical attention [106,107,108,109,110,111,112].

It is essential to note that the likelihood of developing myocarditis or pericarditis post-COVID-19 mRNA vaccination is relatively low, with an incidence of 1 in 200,000 doses for the first dose and 1 in 30,000 doses for the second dose [109,113]. Myocarditis or pericarditis tends to occur more frequently in adolescent males (relative risk of 18.2) and after the administration of the second vaccine dose (relative risk of 15.9) [106]. Moreover, the majority of cases of myocarditis or pericarditis following COVID-19 mRNA vaccination are self-limiting, with recovery observed in 92.6–99.7% of cases without specific treatment [106,108,109,110,112,114].

Presently, there is no consensus or comprehensive research on chest pain in pediatric and adolescent patients following COVID-19 mRNA vaccination. However, given that myocarditis or pericarditis is a high concern for both patients and parents, it is reasonable to perform examination including ECG, serum cardiac enzymes such as troponin I/T, and cardiac ultrasound for adolescents presenting with chest pain after COVID-19 mRNA vaccination in the pediatric emergency department.

## 5. Conclusions

Chest pain in the pediatric population is generally a benign disease, but it often draws parents’ attention and drives them to bring their children to the pediatric emergency department for evaluation [1,2,3,4,7,8]. Several examinations, including ECG, chest radiogram, cardiac troponin, and echocardiography, could be used to approach children and adolescents with chest pain, but most of the diseases could be identified through detailed history-taking and physical examinations. ECGs and chest radiograms had a higher positive finding ratio in patients with associated symptoms and signs than in those with only chest pain [1,47]. Despite high sensitivity and specificity to myocardial injury, cardiac troponin could be elevated in several noncardiac diseases in the pediatric population; therefore, it should be used as a tool when certain cardiac diseases are suspected rather than as a single screening tool for cardiac diseases [4,10,54]. Cardiac POCUS has a high sensitivity and specificity to identify global systolic dysfunction and pericardial effusion, which is helpful for diagnosing pericarditis and myocarditis [21]. A well-designed clinical pathway for pediatric chest pain could decrease unnecessary examination without missing severe diseases presenting with chest pain [3].

## Figures and Tables

**Table 1 diagnostics-14-00526-t001:** Chest pain etiology in different studies.

Study	Year	Population	Region	Sample Size	Cardiac	Respiratory	Musculoskeletal/Idiopathic	Gastrointestinal	Psychogenic	Miscellaneous
Nurul Islam et al. [17]	2023	ED & OPD	India	55	1.8%	1.8%	5.5%	1.8%	83.6%	5.5%
Karabulut et al. [18]	2023	OPD	Turkey	1000	6.8%	7.9%	70.6%	4.1%	7.3%	3.3%
Lubrano et al. [14]	2023	ED	Italy	11,855	4.6%	6.8%	62.5%	8.5%	10.7%	6.9%
Chen et al. [16]	2022	OPD	China	3477	6.7%	10.0%	83.0%	0.29%	0.06%	0.03%
Pissarra et al. [1]	2022	ED	Portuguese	798	1.1%	12.8%	57.5%	5.3%	21.6%	0.7%
Chen et al. [15]	2021	ED & OPD	China	7251	8.0%	9.1%	82.1%	0.6%	0.17%	0.03%
Gastesiet al. [12]	2003	ED	Spain	161	0.6%	8.7%	86.4%	0.6%	0	3.7%
Rowe et al. [9]	1990	ED	Canada	325	1.5%	18.8%	39.1%	0	5.2%	35.1%

ED, emergence department; OPD, outpatient department.

**Table 2 diagnostics-14-00526-t002:** Etiology of cardiac chest pain.

Study	Year	Sample Size	Myocarditis/ Pericarditis n (%)	Structural Heart Disease n (%)	Arrhythmia/ Abnormal ECG n (%)	Cardiomyopathy n (%)	Myocardial Infarction n (%)	Others n (%)
Nurul Islam et al. [17]	2023	55	1 (1.8%)					
Karabulut et al. [18]	2023	1000	7 (0.7%)	11 (1.1%)	27 (2.7%)	2 (0.2%)		21 (2.1%)
Chen et al. [16]	2022	3477	46 (1.3%)		5 (0.14%)	2 (0.05%)		172 (5.9%) *
Pissarra et al. [1]	2022	798	7 (0.9%)	0	2 (0.3%)			
Chen et al. [15]	2021	7251	82 (1.1%)	20 (0.3%)	30 (0.4%)	4 (0.05%)		448 (6.2%) *
Drossner et al. [13]	2011	4288	10 (0.2%)		9 (0.2%)		3 (0.06%)	2 (0.04%)
B H Rowe et al. [9]	1990	325	1 (0.3%)	2 (0.6%)	2 (0.6%)			

Arrhythmia: tachyarrhythmia, bradyarrhythmia, premature atrial/ventricular contraction, Wolff-Parkinson-White syndrome and atrioventricular block; Structural heart disease: atrial septal defection, ventricular septal defect, aortic stenosis, anomalous origin of the coronary artery and pulmonary stenosis; Cardiomyopathy: dilated cardiomyopathy and hypertrophic cardiomyopathy; Others: Suspected myocarditis, mitral valve prolapse, pulmonary embolism, pneumopericardiam, pulmonary hypertension, Marfan syndrome and Kawasaki disease. * Suspected myocarditis was included.

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
