# Peer review of "Pediatric Chest Pain: A Review of Diagnostic Tools in the Pediatric Emergency Department"

_diagnostics, 2024, doi:10.3390/diagnostics14050526_

Round 1

Reviewer 1 Report

Comments and Suggestions for Authors

Thank you for reading this interesting article. I ask the authors to make the following corrections before publication:

1) I believe that the ABSTRACT should be the essence of the results of the analysis. It also lacks conclusions.

2) I don't fully understand the nature of the publication. Too small for a meta-analysis, but quite extensive for a review article. Please consider it after agreeing with the editors on a specific type. However, I propose to construct the work as a review. It should not describe issues as in a meta-analysis of the literature. Rather, please describe specific methods for diagnosing chest pain in children.

3) Taking point 2 into account, I propose changing the title. There is no need for a question formula at the end. It is obvious that appropriate diagnosis must be undertaken. The article only presents the forms of diagnosis and causes of chest pain. The authors do not consider the effectiveness of individual diagnostic techniques.

4) Some of the literature is from the 1990s. You can change them to the current ones or add the latest tests, e.g.:

- Leszczyński P, Wejnarski A, Rzońca P, Gajowniczek A, Gałązkowski R, Mitura K, Sholokhova D. Arrhythmias in children occurring during HEMS intervention: a retrospective cohort study. Emergency Medicine International, 2023. DOI: 10.1155/2023/2974648

(the article shows the latest analysis of heart rhythm disorders in children)

- Krishna PP, Velavarthipati RS, Srikanth M, Krishna BSG, Sriramula N, Goud DPK. To determine the prognostic accuracy of the HEART score as a predictor for major adverse cardiac events in patients presenting with chest pain to emergency department in a tertiary care hospital. Crit. Care Innov. 2023; 6(1): 1-16. DOI: 10.32114/CCI.2023.6.1.1.16

(the article presents the assessment of chest pain cases in the Emergency Department)

Author Response

Response to Reviewers

Firstly, we would like to thank the Editor for giving us the opportunity to revise the manuscript and secondly, express our gratitude to the Reviewers for their valuable comments.

Reviewer #1

1) I believe that the ABSTRACT should be the essence of the results of the analysis. It also lacks conclusions.

Response

    Thank you for your recommendations. We have made significant revisions to the abstract. Additionally, we have added the results of the analysis and conclusion.

2) I don't fully understand the nature of the publication. Too small for a meta-analysis, but quite extensive for a review article. Please consider it after agreeing with the editors on a specific type. However, I propose to construct the work as a review. It should not describe issues as in a meta-analysis of the literature. Rather, please describe specific methods for diagnosing chest pain in children.

Response

    Thank you for your inquiry. The original idea of this article was to focus on a review of “diagnostic tools for pediatric chest pain," which is the fourth section of the manuscript. However, during the writing process, we found it challenging to introduce commonly used diagnostic tools for pediatric chest pain without mentioning its causes. Therefore, we further extended the discussion to include the etiology of pediatric chest pain. As a result, we also supplemented the end of the introduction with an explanation of the genesis of the article, hoping to make the overall structure clearer.

3) Taking point 2 into account, I propose changing the title. There is no need for a question formula at the end. It is obvious that appropriate diagnosis must be undertaken. The article only presents the forms of diagnosis and causes of chest pain. The authors do not consider the effectiveness of individual diagnostic techniques.

Response

    After discussion, we have decided to change the title to "Pediatric Chest Pain: A Review of Diagnostic Tools in Pediatric Emergency Department." This title better captures the theme and is more straightforward.

4) Some of the literature is from the 1990s. You can change them to the current ones or add the latest tests, e.g.:

- Leszczyński P, Wejnarski A, Rzońca P, Gajowniczek A, Gałązkowski R, Mitura K, Sholokhova D. Arrhythmias in children occurring during HEMS intervention: a retrospective cohort study. Emergency Medicine International, 2023. DOI: 10.1155/2023/2974648

(the article shows the latest analysis of heart rhythm disorders in children)

- Krishna PP, Velavarthipati RS, Srikanth M, Krishna BSG, Sriramula N, Goud DPK. To determine the prognostic accuracy of the HEART score as a predictor for major adverse cardiac events in patients presenting with chest pain to emergency department in a tertiary care hospital. Crit. Care Innov. 2023; 6(1): 1-16. DOI: 10.32114/CCI.2023.6.1.1.16

(the article presents the assessment of chest pain cases in the Emergency Department)

Response

    We appreciate your suggestion. We have discussed whether to remove older articles and keep only the newer ones. However, due to the large sample size and clear description of etiology in the study of Rowe et al., we have decided to retain. Additionally, thank for recommending the two articles. Both studies have significantly contributed to the discourse and writing of our article. We have included these two articles in the references and cited them in the "1. Introduction" and "5.1 Discussion" sections to enrich the content and completeness of the article.

Reviewer 2 Report

Comments and Suggestions for Authors

Hunag et al discuss the most common causes of chest pain, both cardiac and non-cardiac, in the pediatric population. They also discuss the main diagnostics methods currently employed to rule out or confirm cardiovascular causes of chest pain. An interesting special adition is represented by the post Covid vaccination myocarditis and pericarditis.

Some minor concerns are listed below.

Please state the type of paper at the beginning of the article (line 1).

The abstract should not be conceived by applying copy and paste from the text body (lines 9-12 are identical to lines 24-28)

There should be a Methods section, explaining how many articles were found in the data base search, how many were included in the analysis, what were the inclusion criteria, and so on.

Line 208: “The diseases caused by cardiac-related chest pain included…”- I think it is the other way around: these diseases caused chest pain.

It would be interesting to insert a subsection of aortic dissection, possible during adolescence in Marfan syndrome patients.

Also please fill in the sections at the bottom of the article: author contribution, funding, etc

Author Response

Response to Reviewers

Firstly, we would like to thank the Editor for giving us the opportunity to revise the manuscript and secondly, express our gratitude to the Reviewers for their valuable comments.

Reviewer #2

1) Please state the type of paper at the beginning of the article (line 1).

Response

Thank you for recommendation; we have added the type of paper.

2) The abstract should not be conceived by applying copy and paste from the text body (lines 9-12 are identical to lines 24-28)

Response

Thank you for your suggestion. We have made significant changes to the abstract, and we have also added the results of the analysis and conclusion.

3) There should be a Methods section, explaining how many articles were found in the data base search, how many were included in the analysis, what were the inclusion criteria, and so on.

Response

Thank you for your reminder. We have added a new section titled "2. Methods," which outlines our approach to searching for articles and the inclusion and exclusion criteria.

4) Line 208: “The diseases caused by cardiac-related chest pain included…”- I think it is the other way around: these diseases caused chest pain.

Response

Thank you for your reminder; the necessary corrections have been made.

5) It would be interesting to insert a subsection of aortic dissection, possible during adolescence in Marfan syndrome patients.

Response

Aortic dissection occurring in adolescents with Marfan syndrome is a noteworthy issue. Therefore, we have reviewed the literatures and added a section titled "5.2 Special consideration: chest pain in adolescents with Marfan syndrome.”

5) Also please fill in the sections at the bottom of the article: author contribution, funding, etc

Response

Thank you for your reminder; the relevant content has been added.

Round 2

Reviewer 1 Report

Comments and Suggestions for Authors

accept